# AI Applied to the Circular Economy: An Approach in the Wastewater Sector

**Vicent Hernández-Chover \*** , **Águeda Bellver-Domingo** , **Lledó Castellet-Viciano and Francesc Hernández-Sancho**

Water Economics Group, Inter-University Institute for Local Development (IILD-WATER), University of Valencia, C/Serpis 29, 46022 Valencia, Spain; agueda.bellver@uv.es (Á.B.-D.); lledo.castellet@uv.es (L.C.-V.); francesc.hernandez@uv.es (F.H.-S.)
**\*** Correspondence: vicent.hernandez@uv.es

**Abstract:** Water is one of the most basic and essential resources for life and is also a strategic component for the development of the economies of the different countries of the planet. The water sector in the context of ecological transition and the circular economy has enormous economic potential. However, the water resources present in a territory are, in many cases, very limited, and their availability is increasingly restricted. In this respect, current technologies make it possible to generate a whole range of renewable resources. In the case of wastewater treatment plants, in addition to obtaining clean water in sufficient quantity and quality, it is possible to take advantage of multiple other resources generated in the purification processes, such as fertilizers, biogas, bioplastics, and glass, and even recover adsorbents such as enzymes and proteins from wastewater. These resources represent a valuable social, environmental, and economic contribution. The scarcity of some of these resources causes continuous increases in market prices, generating economic tensions between producers and potential users. This work proposes to guide the potential of artificial intelligence (AI)-based methodologies in aspects related to the supply and demand of the resources generated in these infrastructures. Specifically, the use of machine learning (ML) allows for projecting economic scenarios based on multiple variables, such as the quality and quantity of the treated flows, the resources generated in the infrastructures, the current demands, and the prices of substitute goods. This aspect represents a substantial advance in terms of the circular economy since, beyond the technical aspects related to the processes, it ensures a sustainable balance between potential producers and end users. In conclusion, it brings sustainability to the urban water-cycle sector, ensuring the viability of the resources generated.

**Keywords:** circular economy; renewable resources; WWTP; machine learning; dynamic pricing; market



## 1. Introduction

In recent years, technological advances have opened up many opportunities. One of the most significant advances has been artificial intelligence (AI), which has made it possible to develop systems capable of performing functions associated with learning, reasoning, and problem-solving. Technological development, together with the ability to analyze and process information, has had positive impacts on the economic, social, and environmental spheres.

From the economic point of view, the use of AI-based methodologies has enabled the analysis of large volumes of data, the continuous monitoring of economic variables in markets, and the integration of different financial systems, increasing the efficiency and privacy of transactions and generating new business models [1]. In the social sphere, AI plays a relevant role in issues related to mobility and the development of cities—for example, it provides sustainable transport solutions that reduce travel times while minimizing the associated environmental impacts, allowing for the greater connectivity of the population and public services. It monitors changes in the population, adapting public services to

these patterns, and, in general, provides solutions that improve the well-being of the population [2]. Finally, at the environmental level, the use of AI makes it possible to monitor biodiversity, optimize the management of natural resources such as water, and enhance their reuse, thus minimizing the impact on the environment. AI is currently used in many sectors to improve decision-making in the management and operation of processes [3]. For example, in the agricultural sector, sensors are used to monitor variables affecting crops, thereby optimizing food production [4]; increase the accuracy of fertilizer, pesticide, and herbicide application [5]; determine optimal planting dates for crops [6]; and help identify and eliminate weeds [7,8]. In the fisheries sector, satellite systems based on optical sensors provide numerous physical parameters of the water, allowing the location of fishing vessels to be determined and underwater drones to determine optimal areas for fishing, and other devices analyze the volume and biophysical characteristics of the fish caught, providing a wealth of information on the supply chain [9]. Similarly, in the livestock sector, data collected by sensors (such as cameras, microphones, accelerometers, gas analyzers...) on animals or their environment, together with sophisticated analytical techniques, provide effective tools to monitor animals in order to improve their welfare and optimize the use of resources, such as feed, water, and land [10]. Some tasks are even performed by robots, such as milking cattle [11] or automatic feed dispensers [12]. Beyond the aspects related to production management, AI allows thousands of variables of different types to be analyzed in a very short time in order to find patterns or trends. These variables can include aspects related to the market, allowing companies to adapt their product introduction or market growth strategies, thus optimizing the profits obtained. Along these lines, there are numerous examples of using AI to optimize marketing channels. In the financial sector, methodologies such as machine learning (ML) are used as tools for fraud detection, investment optimization, automated credit underwriting and decision-making, supplier communication, inflation prediction, liquidity problem detection, or customer identity verification [13–16]. The use of AI allows firms to optimize aspects related to the demand and supply of products and services. Specifically, it allows firms to adapt the price of the product and/or service to specific characteristics related to the market, such as the price of substitute products, total quantities offered, and required or geographical variables. In this way, companies can offer more personalized, tailored, and targeted offers, managing to adjust and individualize their offer by setting a break-even price [17]. The dynamicity provided by market-driven machine learning (ML) allows for real-time data processing, ranking variables and characterizing the offer by evaluating the history, adapting it to the user and thus maximizing the benefits of commercial relationships for the agents involved [18]. However, a large number of examples of the use of AI are based on a constantly growing economy, a linear economy based on extracts, produce, and waste.

It is well known that the scarcity of natural resources can seriously jeopardize the global economy. Since the 1970s, the global extraction of these resources has tripled, and it is expected to double to 190 billion tons by 2060 [19]. The growing demand for critical minerals for future technologies raises concerns about the security of supply, especially given the energy transition from fossil fuels to renewable sources. Today, the levels of exploitation of natural resources have led the European Commission to develop guidelines to alleviate pressure on these resources [20]. These EU-driven guidelines focus on transforming the current economic system from a linear model to a circular one, in which resources and materials are kept in the production system for longer periods of time. However, for this transformation to have a real impact, it is necessary to implement specific measures aimed at optimizing the use of natural resources, while creating new opportunities to manage the waste produced by both industry and society. In this context, the implementation of technological resources based on digitization and the subsequent exploitation of the data generated makes it possible to monitor these developments, facilitating the progress of society in the framework of a circular economy (CE) [21]. The role of AI and its potential in the framework of the CE is particularly relevant in the context of the water sector. The current rise of digitalization is reflected in the Circular Economy Action Plan and the

European Green Pact, adopted by the European Commission in 2020 [22]. These plans have the fundamental objective of building a digital sector oriented toward sustainability and green growth, whereby reinforcing the potentiality of AI in relation to the market possibilities, in terms of the reuse of generated resources, can represent a breakthrough.

On this last point, the urban water cycle can be seen as a strategic sector. Wastewater treatment plants (WWTPs) are capable of generating, in addition to water in quantity and quality, a whole series of resources for other uses. These infrastructures can recover resources from wastewater, such as nutrients (nitrogen and phosphorus) for the subsequent production of fertilizers, generating biogas and energy from organic waste, producing bioplastics, construction materials (glass), and even recovering adsorbents, enzymes, and proteins from wastewater [23]. The use of this alternative waste as a resource in other sectors reduces the pressure on scarce natural resources and would therefore move us toward sustainability and the real implementation of the circular economy in the sector.

Some examples of AI application to productive sectors (agriculture, fisheries, and livestock) have been presented. More specifically, in the wastewater treatment sector, the use of AI has been mainly oriented toward process management: water quality prediction [24,25], the optimization of energy consumption [26], the prediction of BOD and COD [27], and the removal of organic pollutants and micropollutants [28]. However, there are no studies addressing the potential of AI in terms of pricing and monitoring in the wastewater treatment sector. This is because, currently, the resources generated in these infrastructures (water, sludge, energy, fertilizers) are offered to the market following a tariff structure that only includes the costs generated to obtain them. In this sense, taking advantage of the benefits AI has demonstrated in other sectors, such as the financial sector [29,30], can help managers of these infrastructures optimize issues related to the setting and adaptation of prices for the resources offered.

This paper aims to answer the following question: How could artificial intelligence be used to optimize business management and boost demand for the resources generated in the water reuse sector?

The use of AI, oriented to the commercialization of the resources that the water reuse sector is capable of generating, allows for the adjustment of supply to demand in order to incentivize their use and achieve a more sustainable society. This study evaluates the variables to be taken into account and their interactions in the generation of the price of the resources generated, in order to incentivize their demand. This last aspect has hardly been developed in the water reuse sector. Taking advantage of the potential of AI to direct the waste generated to other industrial sectors would make it possible to boost reuse and extend the useful life of resources in general. The goal is to ensure the sustainability of these infrastructures within the framework of the economic development provided by the circular economy.

## 2. Potential for the Use of AI/ML Targeted at Resources Generated in the Wastewater Treatment Sector

Wastewater treatment plants (WWTPs) can recover resources such as bioplastics, enzymes, nutrients, and energy from wastewater, thus creating feedback loops of materials for other uses and at the same time improving the quality of the water discharged [31]. In addition to providing sustainability, these resources can be generated on a continuous basis and can replace some of the conventional resources. For example, in the case of phosphorus, in May 2014, the European Commission classified it as a critical raw material, so alternative sources have since gained importance [32,33]. Recent estimates of phosphorus consumption show that by using available technologies, we can recover up to 30% of the minerals currently used in agriculture by 2030 [34]. Nitrogen is another nutrient of high importance in agriculture; as with phosphorus, it can be recovered from wastewater, thus providing an inexhaustible raw material. Among other examples, the resource par excellence that WWTPs are capable of generating is water, which, if properly treated, can be regenerated until it reaches the quality criteria required according to its use. In the current

context of climate change, water reclamation makes it possible to have one's own water resources, providing self-sufficiency, reliability, and a guarantee of supply.

To this end, it is important to have forecasting techniques to strengthen the distribution channels that link the supply of these resources with their demand, to ensure that they are used efficiently and sustainably. Distribution and marketing channels are the links between producers and consumers; for the supply of resources, they are defined by the WWTPs, and for the demand for resources, they are defined by the potential users.

In the wastewater treatment sector, we can find different user profiles. In general, three main groups can be defined: the agricultural sector, the industrial sector, and the domestic sector. With regard to the demand for resources, the agricultural sector mainly requires water in sufficient quantity and quality for crop irrigation; sludge for soil fertilization; and fertilizers such as nitrogen and phosphorus. In the case of the industrial sector, there are numerous industries that require a high quantity of water in their production processes, and depending on the industrial process they carry out, they may need water of different qualities. In the case of the domestic sector, industries may mainly require water for garden irrigation, the maintenance of ornamental fountains, street cleaning, or water storage tanks for firefighting and electricity to satisfy energy consumption of various kinds [35].

In the distribution and marketing channels, the interaction between suppliers and demanders requires a certain balance that guarantees the economic relationship between the parties. The main variables that intervene are the quantities offered of each resource, its availability, the price of the resource offered, and the prices of current substitute goods, in addition to other aspects. Price plays an incentive role in trade, so achieving an equilibrium price implies maximizing utility for the agents involved in the market. Some of the possible applications aimed at optimizing the supply and demand of the resources generated in the urban water cycle are presented below.

### 2.1. Water Distribution and Commercialization

Ensuring an adequate quantity and quality of water supply is a challenge in many areas of the world due to factors such as climate change, water pollution, and rapid growth in water consumption. In this regard, as industrialization increases in developing countries, industrial water use could intensify, putting water resources under severe stress. At present, the application of various physicochemical treatments allows wastewater to be used for other purposes. The latest studies show how current technological development makes it possible to adapt reclaimed water to different purposes, and it can be used directly for uses that require stricter quality criteria or for other processes subject to lower quality requirements [36]. In short, the use of reclaimed water frees up resources of better quality for uses that are more restrictive in relation to the quality required. Connecting the supply of reclaimed water with the demand makes it possible to adapt the quality of the wastewater to the required criteria, thus making it possible to offer different qualities. Distributing different qualities of water implies having different piping networks in order to ensure its quality, so the costs of producing different qualities will vary depending on the investment required in the technology and infrastructure necessary for its distribution, as well as the operating and maintenance costs of these facilities. Establishing an equilibrium price also implies taking into account other variables such as the current price that users pay for the conventional resource, as well as its availability. This criterion can act as an incentive for the demand for non-conventional resources due to the fact that, in certain regions, water shortages lead to service stoppages and price increases.

The use of ML can be of great help in setting the prices of the different qualities of water offered. It allows us to combine the history of the volumes treated in the infrastructure and the costs associated with the purification and regeneration processes, and, in addition, to take into account the quantities to be treated according to the existing demand. Knowing the aspects related to the variability of demand, prices, and the availability of conventional resources allows for maximizing the benefits of the WWTP while guarantee-

ing the availability of water, whether for agricultural irrigation or industrial or domestic processes [37].

### 2.2. Distribution and Commercialization of Nitrogen and Phosphorus

Nitrogen and phosphorus are two important nutrients that are present in wastewater; if released in excess in natural water bodies, they can cause environmental problems such as eutrophication, which is the excessive enrichment of nutrients in water [38]. In the agricultural sector, phosphorus and nitrogen are the main nutrients used as fertilizers; in this sense, the UN calls for measures to better use this fertilizer and produce it in a sustainable way, extracting it from wastewater. According to the UN, phosphate rock is the main source of phosphorus worldwide for the production of synthetic fertilizers. However, its availability is increasingly limited, and with a growing population, it may be insufficient to meet agricultural and, ultimately, food demands [39,40].

Sewage treatment plants are capable of extracting both minerals from struvite, a mineral with high concentrations of nitrogen and phosphorus, as well as magnesium. Many countries already use it as a fertilizer for crop fields because of its advantages (slow dissolution), as well as the fact that it reduces the risk of water contamination [41].

If we look at the current fertilizer market, international reference prices increased throughout 2022, and many quotations reached all-time highs. Fertilizer prices are determined by the interaction of supply and demand. Different causes have generated this relentless price increase—on the supply side, these include (i) high and increasing energy prices and (ii) trade disruptions and high transportation costs, while on the demand side, they include (iii) high crop prices (and therefore, high affordability).

The use of ML allows multiple variables to be included in order to establish a dynamic price and synchronize aspects such as daily quotations of industrial fertilizers, consequently allowing us to adjust the price of natural fertilizers produced in WWTPs in order to optimize their distribution at the farm level. This information can be defined at earlier stages of the wastewater treatment process. Knowing the characteristics and volume of water entering the plant makes it possible to project the quantities of fertilizers that the WWTPs can offer to the market, making it possible to plan and supply the demand to them while minimizing the impacts generated by the extraction and subsequent transformation of industrial fertilizers.

### 2.3. Sludge Distribution and Commercialization

Sludge generated at WWTPs is a solid by-product of wastewater treatment. These sludges, also known as sludge or biosolids, are the result of the removal of solids and organic matter during the water purification process. They are widely used as agricultural soil fertilizers [42,43].

In terms of agricultural demands, these sludges are often used as organic fertilizers in agriculture. They contain valuable nutrients such as nitrogen, phosphorus, and organic matter that can improve soil quality and promote crop growth. Their use must be carried out following guidelines and regulations, such as considering the quality and composition of the sludge to determine the appropriate amount to apply to agricultural fields. In the agricultural sector, it contributes to alleviating the health of soils deficient in organic matter, improving their characteristics and providing them with a greater capacity to protect themselves from erosion, and, ultimately, desertification [44,45].

In this case, the use of ML makes it possible to optimize the sludge distribution channel. There are different variables that can be monitored to automate shipments, adjusting quantities and qualities to the agricultural terrain. In addition, geographic localization can be of great help to optimize transport routes, minimizing total transport times and, consequently, economic costs and associated emissions. Finally, the economic prices associated with sludge EUR/tn.) can vary and adapt to market conditions, so a higher demand for sludge can generate slight price increases and vice versa, among other aspects.

### 2.4. Energy Distribution and Commercialization

Generating electrical power at a wastewater treatment plant (WWTP) is a practice that can be beneficial both from an environmental and economic point of view. For this purpose, there are different technologies for which the choice depends on several factors, such as the availability of resources (such as treatment sludge), wastewater flow, geographic location, and the plant's sustainability and energy efficiency objectives [46]. In general, the energy generated in a WWTP comes from renewable sources, such as anaerobic sludge digestion and biogas capture. Once generated, there are different distribution options: self-consumption, distribution to nearby facilities, storage, grid injection, heating, steam generation, or selling renewable energy credits.

Each of the possible alternatives involves a set of variables that must be analyzed on an ongoing basis; this is mainly due to the current fluctuation of market prices. However, some of the proposed alternatives may pose higher investment costs that must be addressed—for example, the accumulation of the energy generated requires batteries that allow it to be used at specific times. Likewise, self-supply must consider times when the energy consumed may exceed the energy produced, so it is important to take into account the restrictions that each alternative may pose. However, methodologies such as ML can help to make daily decisions in order to optimize how the infrastructure uses the energy; thus, we would be maximizing the benefits that producing energy can generate for the WWTP, taking into account the price of conventional energy [47–50].

In conclusion, AI/ML allows us to optimize decisions in the distribution and commercialization channels. It manages to automate decisions by considering a multitude of dynamic parameters that can be processed and evaluated almost instantaneously, in order to adapt the decisions or prices of the resources generated to the market. For this, in addition to the variables that define the market, it is necessary to have data relating to certain technical aspects, as well as the history of the plant in terms of flows treated and resources generated. This last aspect allows us to reduce uncertainty because we are able to make forecasts of both production and demand for the resources generated.

## 3. Machine Learning Applied to the Wastewater Treatment Sector

The wastewater treatment sector has great potential in terms of resource generation. It is considered a strategic sector due to the large amount of resources it produces, and it guarantees economic development within the framework of sustainability and the circular economy. The use of methodologies that monitor the production of the resources generated in the WWTP allows, beyond the management of processes, for the optimization of marketing and distribution channels, adjusting the demand and supply of these resources. In this last aspect, the design of appropriate tariffs will make it possible to incentivize the demand for these resources. In terms of their design, there are numerous aspects involved, which can be divided into internal factors—costs intrinsic to their production—and external factors—unique characteristics of the market that affect their demand. A constantly changing environment implies a continuous adaptation of the prices offered for the different resources that these infrastructures are capable of generating.

In a competitive market, consumers can choose between different goods/products to satisfy their needs. The concept of substitute goods refers to goods that compete with each other because they can perform the same function or service. For example, in the case of the water resource, it is common for both industrial and agricultural areas to resort to conventional fresh water supply, either through installations carried out by the drinking water supply company or through the direct extraction of water from aquifers, rivers, etc. However, WWTPs are capable of providing water in sufficient quantity and quality, adapting to the requirements of the industrial and agricultural sectors. In this example, there are two assets that can be used in the same way; one of them is conventional (limited natural resource), and the other is non-conventional (through regeneration). With regard to the consumption of water from the WWTP (non-conventional resource), it may depend on characteristics related to geographical areas. For example, areas with high water stress will

be encouraged to use non-conventional water due to the possible limitations posed by the conventional resource. Between the two most polarized scenarios (situations of water stress and water surplus), there are multiple scenarios in which factors such as the price and availability of the resource will play a determining role in the final choice of the resource.

Similarly, in the agricultural sector, sludge and fertilizers are used with the aim of increasing crop productivity and improving soil characteristics. With regard to the use of these resources, WWTPs are capable of generating both manure and fertilizers (phosphorus and nitrogen). However, in the latter case, it is common to use industrial fertilizers, the availability of which is becoming increasingly limited. In a market with scarce resources, prices can suffer upward variations and, as in the case of fertilizers, costs have been rising in recent years, putting the food industry at serious risk. In this scenario, fertilizers obtained through sustainable processes, such as their recovery from wastewater, guarantee the long-term sustainability of the sector. However, the price of this resource may be affected by the prices of industrial fertilizers (substitute goods), so a shortage in the production of the latter would increase the consumption of fertilizers produced in WWTPs.

Finally, energy represents another strategic resource and, likewise, there is a range of alternatives with which to guarantee energy demand, whether domestic or industrial. In recent years, the prices associated with energy have experienced continuous growth, so industries that depend heavily on energy to carry out their production processes have seen an alarming increase in production costs. Furthermore, it is important to point out that the source of the energy can generate high environmental impacts and pose certain risks, mainly due to the process of obtaining it. In this scenario, having a clean, alternative source of energy, such as the WWTP, would guarantee the sustainability of nearby industries and agricultural fields (pumping water for irrigation) or the supply of this infrastructure, avoiding the consumption of other external energy sources.

As can be seen in the following figure (Figure 1), the existence of both internal and external factors can influence the demand for this type of resource. Therefore, methodologies are required to adapt both the quantities offered and the appropriate prices to the different users (industrial, agricultural, and domestic sectors). Knowing these variables in advance would make it possible to adjust a tariff to adapt to a constantly changing environment.

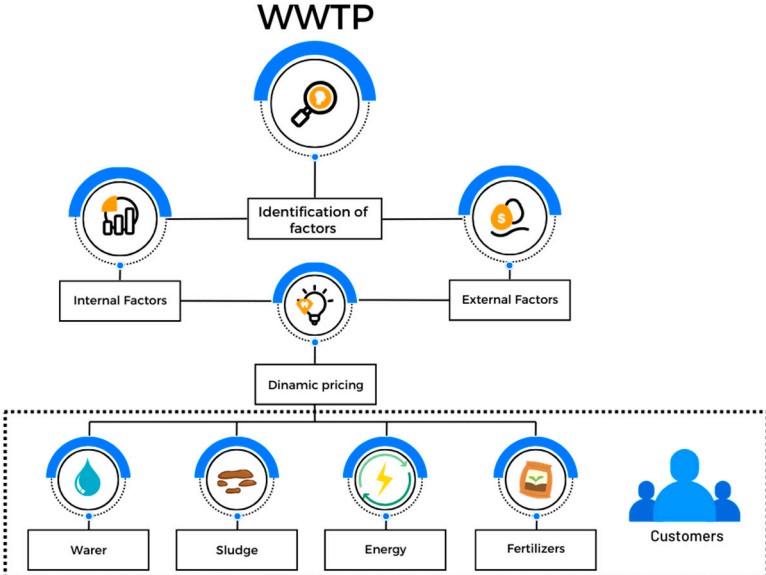

**Figure 1.** Diagram of the process and resources generated.

Advances in machine learning, in terms of price prediction, provide algorithms capable of processing large amounts of data. Aspects related to both production costs and external variables are related to the market in which they operate, so they can be used to solve pricing problems [46]. One example is the airline industry [51], where ML is applied as a pricing

system for airlines. They use AI to provide a dynamic pricing model that continuously adapts to market circumstances. Their simulation includes real-world problems, e.g., overbooking and the existence of different customer classes.

The price of the different resources generated in WWTPs can be approximated in two ways. Price prediction using a regressor model [52] is generally applied to result in a numerical value. The other approach is the ranking model [53], which cannot predict prices as numerical values but can make decisions on whether to increase or not. In the latter case, it is possible to apply the decision tree to the wastewater treatment sector, with the aim of suggesting percentage increases in the prices offered for each resource generated.

### 3.1. Empirical Case: Applying ML to the Wastewater Treatment Sector

An important knowledge structure that can result from data mining activities is decision trees. They can be used for the classification of future events, and these events can be defined by both qualitative and quantitative criteria. The present work applies the decision tree methodology with the objective of integrating a large number of variables that can influence price.

Decision trees are classified within nonparametric supervised learning methods and can be used for both classification and regression. The purpose is to create a model that is able to predict the value of a target variable through simple decision rules inferred from the characteristics of the data [54]. Decision trees are generally constructed recursively, following a top-down approach. A standard decision tree consists of a number of branches, a root, a number of nodes, and a number of leaves. A branch is a chain of nodes from the root to a leaf, and each node implies an attribute. The occurrence of an attribute in a tree provides information about the importance of the associated attribute. Next, the algorithm splits the data using all possible binary splits and chooses the one that by splitting the data minimizes the sum of the squared deviations from the mean in the new branches. The splitting process continues to be applied to each created branch until each node reaches a minimum node size and becomes a terminal node. This minimum size is specified by the user through the number of training samples at the node [55]. The procedure for forming the decision tree and exploiting it for pricing is characterized by:

- The set of statistical characteristics can come from history, e.g., reclaimed water prices in recent years; these can be defined by infrastructure characteristics and specific market conditions in which the company operates.
- The decision tree has leaf nodes, which represent class labels, and other nodes, which are associated with the classes (magnitude level in this case) being analyzed.
- The branches of the tree represent each possible value of the parameter node from which they originate.
- The decision tree can be used to express the structural information present in the data by starting at the root of the tree (the highest node) and moving through a branch to a leaf node.
- The level of contribution of each individual parameter is given by a statistical measure within the parentheses in the decision tree. The first number in parentheses indicates the number of data points that can be classified using that set of parameters. The parameters appearing in the decision tree nodes are in descending order of importance.
- At each decision node, the most useful parameter for classification can be selected using the appropriate estimation criteria. The criterion used to identify the best parameter invokes the concept of entropy and information gain, which is discussed in detail in the following subsections. The decision tree algorithm has two phases: construction and pruning. The construction phase is also known as the "growth phase".

The algorithms used to build decision trees are several, CART (classification and regression), C4.5, CHAID, and QUEST [56]. The following table (Table 1) summarizes the four most commonly used algorithms [57,58].

**Table 1.** Comparison of different decision tree algorithms.

| Methods | CART | C4. 5 | CHAID | QUEST |
|---|---|---|---|---|
| Measure used to select input variable | Gini index; Twoing criteria | Entropy info-gain | Chi-square | Chi-square for categorical variables; J-way ANOVA for continuous/ordinal variables |
| Pruning | Pre-pruning using a single-pass algorithm | Pre-pruning using a single-pass algorithm | Pre-pruning using Chi-square test for independence | Post-pruning |
| Dependent variable | Categorical/Continuous | Categorical/Continuous | Categorical | Categorical |
| Input variables | Categorical/Continuous | Categorical/Continuous | Categorical/Continuous | Categorical/Continuous |
| Split at each node | Binary; Split on linear combinations | Multiple | Multiple | Binary; Split on linear combinations |

Algorithms require data for their operation; in fact, a greater availability of all historical data allows us to calibrate the model, thus improving the results. In addition, they allow us to include both categorical and continuous variables, so that we can enrich the results with dichotomous variables that respond to more qualitative aspects of the processes. Discretizing allows continuous variables to be converted into categories or ranges. This helps to simplify the rules that the decision tree uses to make predictions. Discretization makes decision trees more effective, interpretable, and efficient, which facilitates data-driven decision-making. The target variable (increase the price of unconventional water) can be described as a discrete variable, taking values between 1 and 10.

These values can respond to trends, taking as reference quarters or semesters over time. Thus, a continuous increase in the price of conventional water can be classified according to the values described. The variables can be normalized in order to always express them on a scale of 1 to 10, taking a minimum and a maximum (Table 2).

**Table 2.** Behavior of variables. Temporal increases or decreases (%).

| | Month 1 | Month 2 | Month 3 | Month 4 | Average |
|---|---|---|---|---|---|
| Variable 1 | 7 | 6 | 2 | 2 | 4.25 |
| Variable 2 | 2 | 6 | 7 | 5 | 5.00 |
| Variable 3 | 7 | 4 | 2 | 3 | 4.00 |
| Variable 4 | 3 | 3 | 4 | 5 | 3.75 |
| Variable 5 | 2 | 7 | 7 | 7 | 5.75 |
| Variable 6 | 3 | 3 | 2 | 4 | 3.00 |

The average obtained can then be interpreted as follows: a value close to 10 suggests an increase in price and a value of 1 suggests price maintenance (Table 3). The set of results obtained according to the chosen variables will provide an overall result (target variable) that will help to make decisions on possible increases in the price of the resource.

The variables that can influence the selling price of the example resource (non-conventional water) are then identified and discretized. The variables are divided into internal and external factors; this is due to the nature of the characteristics, with some depending on the internal functioning of the infrastructure and others on market-related aspects.

**Table 3.** Discretization of results.

| Results (Average of Variables) | Target Variable | Action |
|:---:|:---:|:---:|
| 0–1 | 1 | |
| 1–2 | 2 | |
| 2–3 | 3 | Maintain price |
| 3–4 | 4 | |
| 4–5 | 5 | |
| 5–6 | 6 | |
| 6–7 | 7 | |
| 7–8 | 8 | Increase price |
| 8–9 | 9 | |
| 9–10 | 10 | |

3.1.1. Internal Factors

The internal factors are defined by all variables related to the management and production of the infrastructure. First of all, in order to estimate market prices, it is necessary to know both the investment costs (CAPEX) and the treatment and operating costs (OPEX) of all the processes involved in obtaining the resources (reclaimed water, sludge, fertilizers, energy, etc.). In this case, we take as an example the non-conventional water that WWTPs can offer for other uses (industrial or agricultural). It is important to guarantee the recovery of all costs to ensure the sustainability of the process.

In the case of reclaimed water, a fundamental aspect in determining costs is the type of technology used in the treatment [58,59], which will depend on the quality standards required for its use established in Royal Decree 1620/2007, of December 7 (RD 1620/2007), in Spain. The reuse project should provide for the amortization of all investment costs, and this aspect is closely related to the different water qualities that the WWTP can offer in nearby areas. For example, in the industrial sector, it is possible to use two qualities: (A) low-quality reclaimed water that can be used in the cleaning processes of equipment, facilities, and processes that do not require high water quality; and (B) higher-quality water obtained from advanced tertiary treatment, which can be used in processes that require high-quality water. In addition to costs, there are other aspects that must be considered. As in any other industry, there are variables that can affect production; in this case, they must be monitored in order to know the quantities that can be offered to consumers—in our case, in the agricultural and industrial sectors. The quality of the influent (measured by the concentration of COD, BOD, suspended solids, nitrogen, and phosphorus) together with the possible fluctuations of the inflow can generate some variability in the quantities that can be offered to the market. The following figure (Figure 2) visually summarizes the different internal aspects to be considered.

(a)　Capital and Operating Expenditures

The economic costs associated with investment in machinery or infrastructure and operation (reagents, personnel, electricity) must be recovered. They are usually expressed in monetary units per $m^3$ produced (EUR/$m^3$). An increase in these costs implies a possible increase in the final sales prices offered to users. Daily monitoring is important because some costs, mainly operating costs, can vary significantly, as is the case with electricity (Table 4).

(b)　Daily Flow

Wastewater can fluctuate with increasing or decreasing use, both in households and in industry. In addition, rainfall can affect the flow, increasing the volume to be treated. Knowing the daily flow that enters the wastewater treatment plant allows us to adapt the purification and regeneration processes, as well as know the volumes of water that we will be able to offer to users (Table 5).

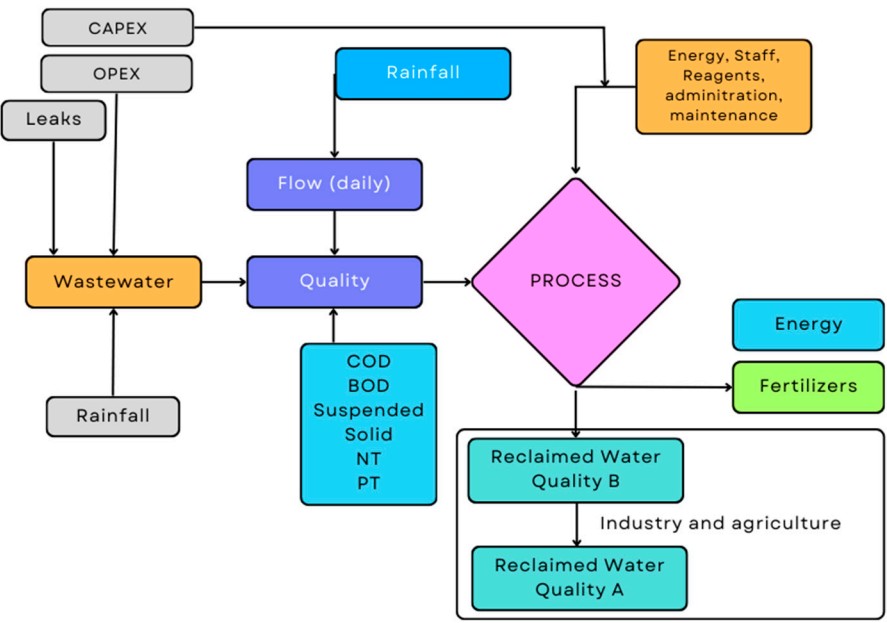

**Figure 2.** Flowchart internal aspects.

**Table 4.** Percentage change in costs and result (Discretization).

| 0–1% | 1–2% | 2–3% | 3–4% | 4–5% | 5–6% | 6–7% | 7–8% | 8–9% | 9–10% |
|------|------|------|------|------|------|------|------|------|-------|
| 1 | 2 | 3 | 4 | 5 | 6 | 7 | 8 | 9 | 10 |

**Table 5.** Daily flow reduction (Discretization).

| −(0–1%) | −(1–2%) | −(2–3%) | −(3–4%) | −(4–5%) | −(5–6%) | −(7–8%) | −(8–9%) | −(9–10%) | −(10–11%) |
|---------|---------|---------|---------|---------|---------|---------|---------|----------|-----------|
| 1 | 2 | 3 | 4 | 5 | 6 | 7 | 8 | 9 | 10 |

Following Table 5, a decrease in the daily treated flow will suggest an increase in price; this assumption rests on the availability of water supply for both quality A and B.

(c)    Quality of the Influent (Organic Load)

The monitoring of basic wastewater parameters such as biochemical oxygen demand (BOD), chemical oxygen demand (COD), suspended solids (SS), nitrogen (N), and phosphorus (P) is of relevant importance. The costs of both purification and reclamation processes depend to a large extent on the higher or lower concentration of pollutants in the incoming water. Water with a higher concentration of pollutants will require a higher use of reagents and energy. Following the explanation above, an increase in the concentration of contaminants will suggest to the operator an increase in the final price offered (Table 6).

**Table 6.** Increase in the average concentration of pollutants in wastewater (Discretization).

| 0–1% | 1–2% | 2–3% | 3–4% | 4–5% | 5–6% | 6–7% | 7–8% | 8–9% | 9–10% |
|------|------|------|------|------|------|------|------|------|-------|
| 1 | 2 | 3 | 4 | 5 | 6 | 7 | 8 | 9 | 10 |

Another consequence directly related to the quality of the incoming wastewater is the amount of resources that can be generated. For example, the amount of fertilizer depends directly on the concentration of nitrogen and phosphorus in the wastewater.

(d)    Precipitation in the Area

In many areas, there is no separative network, so rainfall implies a higher flow to be treated by the WWTPs. Knowing the average rainfall, as well as the daily history, makes it possible to adapt the infrastructure processes and, consequently, to know in greater detail the peak flows that can be distributed (Table 7).

**Table 7.** Percentage reduction in precipitation (1mm). (Discretization).

| −(0–1%) | −(1–2%) | −(2–3%) | −(3–4%) | −(4–5%) | −(5–6%) | −(7–8%) | −(8–9%) | −(9–10%) | −(10–11%) |
|---|---|---|---|---|---|---|---|---|---|
| 1 | 2 | 3 | 4 | 5 | 6 | 7 | 8 | 9 | 10 |

A decrease in rainfall would imply a reduction in the flow to be treated and therefore less non-conventional water. Reduced availability would justify an increase in the prices offered.

(e)　Leaks (Sanitation Network)

Leaks in the sanitation network can be a serious problem that affects the efficiency of the system and the environment. The volume of wastewater lost implies less waste generation at the WWTP. Knowing the losses in detail helps to identify the real capacity of waste generation and, consequently, to plan interventions in the network in order to increase efficiency. Establishing an investment plan to reduce wastewater losses may imply an increase in investment (CAPEX); consequently, the prices offered for non-conventional water will have to increase in order to cope with the investment and consequent amortization. This variable can be difficult to quantify because it depends on the total investment to be made; however, the influence of the investment on the final price offered can follow the pattern of the tables above to suggest to the operator a final increase. Losses can be measured per $m^3$ per kilometer of the network. The difference is defined by the water registered at the meters and the wastewater received at the wastewater treatment plant. In cases where the network is sectorized, it is relatively simple to calculate the percentage of the network to be renewed (Table 8).

**Table 8.** Percentage of network to be renovated (Discretization).

| 0–1% | 1–2% | 2–3% | 3–4% | 4–5% | 5–6% | 6–7% | 7–8% | 8–9% | 9–10% |
|---|---|---|---|---|---|---|---|---|---|
| 1 | 2 | 3 | 4 | 5 | 6 | 7 | 8 | 9 | 10 |

(f)　Quantities of Reclaimed Water to be Produced:

It is important to know at present the consumption of conventional water in the different sectors, mainly agricultural and industrial. In addition, the quality required for their processes will justify the use of a particular technology. An increase in the quantity demanded, whether of quality A or B, will justify an increase in price (EUR/$m^3$). This is related to the availability of conventional water and the urban, agricultural, and industrial development of the study area. An increase in non-conventional water demand (last 6 months) allows for assigning a 10 to this characteristic. The choice of a six-month period justifies a trend in consumption. This variable is also explained as an external factor because water demand is determined by the users and the study area. In order to avoid duplication, it is quantified as an external factor.

### 3.1.2. External Factors

On the other hand, external aspects are defined by all the variables that can influence the demand for the resource generated, mainly by gathering information from the market. Constant monitoring of aspects related to the price of conventional resources, quality, availability (water scarcity), and parameters related to the current service (possible service interruptions) can influence the final price offered to the market. For example, a decrease

in both the quality and availability of conventional resources will act as an incentive to use reclaimed water. The following illustration summarizes the external aspects to be considered (Figure 3).

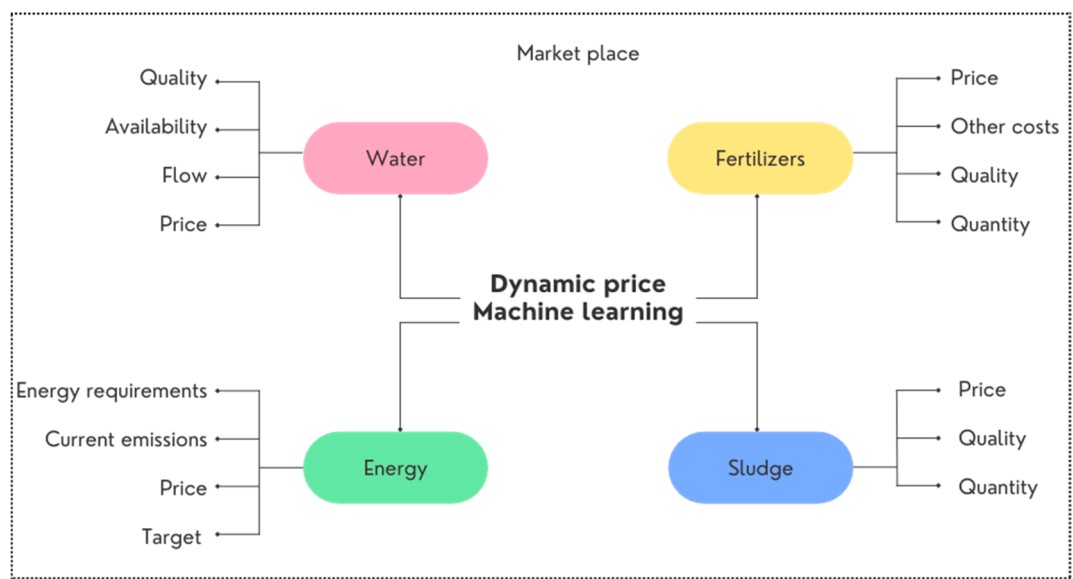

**Figure 3.** Flowchart of external aspects.

(a)   Conventional Water Quality (Aquifer, Network, Surface)

The physical–chemical characteristics of the water must meet minimum requirements for its use. In certain geographical areas, there may be contaminants in the water that prevent it from being used, so this aspect must be taken into account. A decrease in the quality of conventional water implies higher treatment costs, so the price of conventional water will be affected (upward). This aspect is easier to discretize because a slight increase in the price of conventional water will allow for an increase in the price of reclaimed water. The following table (Table 7) shows an average increase in the four main pollutants: chemical oxygen demand (COD), suspended solids (SS), nitrogen (N), and phosphorus (P) (Table 9).

**Table 9.** Increased concentration of pollutants in conventional water (aquifers, rivers) (Discretization).

| 0–1% | 1–2% | 2–3% | 3–4% | 4–5% | 5–6% | 6–7% | 7–8% | 8–9% | 9–10% |
|------|------|------|------|------|------|------|------|------|-------|
| 1 | 2 | 3 | 4 | 5 | 6 | 7 | 8 | 9 | 10 |

(b)   Availability of Conventional Water

This variable must be considered to establish the price of the non-conventional resource; it can be expressed qualitatively—for example, by assigning an order (three or more alternatives) to the availability of the resource (polytomous). A lower availability (water scarcity) will imply a higher price for the conventional resource (Table 10).

**Table 10.** Discretization of conventional water availability.

| Conventional Water Availability | |
|---|---|
| Regular supply disruptions | 10 |
| Sporadic outages | 5 |
| Continuous service | 1 |

The quality of service is an aspect that can be detrimental to industries that use the resource in their production process. For example, cardboard, paper, ceramics, and textile companies need to ensure the availability of water in their manufacturing processes. Thus, if the area suffers from water shortages, the WWTP can guarantee the continuous service of reclaimed water in order to meet the sector's demand.

(c)     Flow Consumed

The quantities demanded by users of both qualities are of utmost importance; knowing them allows for planning the supply of non-conventional water and adapting the price following the principle of profit maximization. An increasing trend implies an increase in the price of the offered resource (non-conventional water); this trend can be observed over time periods corresponding to months, quarters, or semesters. At the same time, the system allows for differentiation between the different qualities offered and is able to vary the price separately for each quality (A or B). For example, a continued increase in flow demand over 6 months may average 3 and 6%, scoring 5 and 10, respectively (Table 11).

**Table 11.** Average flow rates consumed (qualities). Expressed as a percentage increase compared to the previous period (%). Discretization.

|  | Flow Consumed (m$^3$) | |
|  | Quality A | Quality B |
| --- | --- | --- |
| Month 1 | +3 | +6 |
| Month 2 | +5 | +7 |
| Month 3 | +4 | +3 |
| Month 4 | +2 | +4 |
| Month 5 | +4 | +5 |
| Month 6 | +4 | +8 |
| Average | +3.67 | +5.50 |
| Discretization | 5 | 10 |

(d)     Price

In the case of Spain, each municipality designs and sets its own freshwater tariffs. This may cause some differences depending on the geographical area assessed. We understand that non-conventional water is a substitute for freshwater, which can be used for both agricultural and industrial purposes. In this sense, users will normally opt for the resource (water) that offers the lowest price. In this case, the price of freshwater can be a restriction for the consumption of reclaimed water. Therefore, the proposed model must monitor the current freshwater prices to set a maximum reference value for both qualities (A and B). Beyond the maximum price constraint, the water situation in the Mediterranean area may also lead to continuous increases in the price of freshwater. In this situation, an increase in the price of the conventional resource should imply an increase in the non-conventional water resource offered to maximize the profits of the wastewater treatment plants. In the latter case, the variable takes dichotomous values; in the case of an increase in the price of freshwater, the discretized variable takes the value 10.

## 4. Conclusions

This work proposes to direct the potential of methods based on artificial intelligence (AI) toward aspects related to the supply and demand of resources generated in wastewater treatment plants (WWTPs). In the context of the circular economy, these infrastructures play a relevant role as they are able to produce, in addition to non-conventional water, fertilizers and electricity, among other resources. In this sense, WWTPs can be considered a source of production of non-conventional resources, with the capacity to satisfy different needs of productive sectors, such as agriculture and industry. Therefore, they are another economic actor in the market capable of competing with the current supply of resources from non-renewable sources. In this vein, in order to bring economic sustainability to the

wastewater treatment sector, methodologies are needed to help operators optimize the tariffs offered for the resources produced, adapting prices to market conditions.

At present, the prices of these resources, whether reclaimed water or fertilizers, respond to fixed tariffs based solely on the costs incurred in the production processes. Setting sales prices (tariffs) solely on the basis of the costs incurred in the process can reduce the profits earned by the operators of these infrastructures. This is due to the fact that in the market there are a series of fluctuations that can affect the availability of natural resources, generating situations of scarcity that directly affect prices, increasing them. This situation generates a series of opportunities that can be exploited by the wastewater treatment sector. Therefore, new approaches are needed to monitor the market conditions and to adapt the price (supply) to the characteristics of the existing demand. In short, maximizing the profits generated will ultimately ensure the economic sustainability of these facilities.

The paper presents a practical application that uses a machine learning (ML) methodology to monitor and evaluate in real time the change in different variables, providing a dynamic solution in terms of tariff-setting for the resources generated in the wastewater treatment sector. The demonstration includes an empirical case that specifically addresses the marketing and distribution of reclaimed water. It identifies the variables that may influence the final price of the resource and classifies the possible changes they may undergo, addressing both aspects related to the infrastructure itself and those related to the market in which it operates. For ease of understanding, the result is expressed on a scale of 1 to 10, where 10 indicates an increase in the price of the resource. The results obtained include changes related to the availability and scarcity of natural resources, which are identified and taken into account in order to optimize the tariffs of unconventional resources produced in these facilities. This dynamic requires the use of methodologies capable of monitoring and evaluating in real time the impact of multiple variables on price generation.

The possibility of automating in real time the variables that influence prices represents a breakthrough in terms of the circular economy; beyond the relevance of the water resource, it is possible to supply society with other resources that are strategic for economic development. However, this approach can also have certain limitations: first of all, it is necessary to automate the process of obtaining data, both technical and economic. In this sense, the provision of results in real time depends on the updating process. Secondly, some external factors, such as the availability of the conventional resource to be evaluated, suggest the definition and discretization of the possible results, so some variables may be subject to the environment in which they operate and require a certain consensus that represents the reality of the market. Finally, there are certain social barriers that must be taken into account. The demand for these resources may represent a certain rejection due to the fact that they do not come from conventional sources.

This paper concludes by highlighting the strategic role that wastewater treatment plants play in the production of unconventional resources. The dynamic circumstances of the market generate commercial opportunities that operators can take advantage of. Having tools with which to update prices based on market characteristics makes it possible to maximize the profits obtained, thus ensuring the sustainability of the wastewater sector within the framework of the circular economy. The analysis of the different possibilities offered by the use of ML in the marketing channel serves as an illustrative and stimulating example for many companies in the sector, offering a point of view that can enrich the possibilities offered by AI.

**Author Contributions:** Conceptualization, V.H.-C.; Methodology, V.H.-C.; Formal analysis, Á.B.-D. and L.C.-V.; Writing—original draft, V.H.-C. and Á.B.-D.; Writing—review & editing, F.H.-S. All authors have read and agreed to the published version of the manuscript.

**Funding:** Valencian Government (project CIAICO/2021/347-Generalitat Valenciana) and Spanish Government (project MRR/TED2021-132872B-I00).

**Institutional Review Board Statement:** Not applicable.

**Informed Consent Statement:** Not applicable.

**Data Availability Statement:** Data are contained within the article.

**Conflicts of Interest:** The authors declare no conflicts of interest.

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
