# Peer review of "AI Applied to the Circular Economy: An Approach in the Wastewater Sector"

_sustainability, doi:10.3390/su16041365_

Round 1

Reviewer 1 Report

Comments and Suggestions for Authors

The study addresses the up-to-date topic of applying artificial intelligence technologies to transition to circular economy principles in the field of water management. However, the article is very poorly structured. A lot of unnecessary information not directly related to the goals and objectives of the study. The research methodology is poorly described. As a result, the authors' contribution and the novelty of their research are not clear. 

The introductory section is poorly organised. It is good to express the need for the study with the backlog of literature that exists in the framework. There is no background in this introduction stating the urgency and novelty of the study where innovative ideas need to be flown through the background along with the useful insights. What is the novelty of this article compared to existing studies? What is the scientific contribution of your work to science? Can you, please, indicate the scientific implications of your paper?The structure of the article does not conform to the generally accepted structure: there is no literature review section, no section devoted to research methodology. The statement of research questions is not clear.

Reviewer 2 Report

Comments and Suggestions for Authors

1. The authors need to add a description of the methods used in this study

2. The authors need to add an analysis of empirical data to support their conclusions

Comments on the Quality of English Language

No comments

Reviewer 3 Report

Comments and Suggestions for Authors

This work proposes to guide the potential of artificial intelligence (AI)-based methodologies in aspects related to the supply and demand of the resources generated in these infrastructures. Add a research question before the objective.

Currently, it is important to evaluate AI in interface with other areas of knowledge, e.g., CE. Bring to the manuscript new theoretical and practical contributions, critical discussion, trends and opportunities in the field of AI and CE in the wastewater sector.

The abstract extensively contextualizes the topic, whereas the focus should center on practical outcomes derived from the application of AI within the circular economy. Specifically, the narrative needs to emphasize tangible results stemming from AI's implementation within this context.

Regarding the methodology section, it lacks clarity in structuring. It would greatly benefit from introducing a clear division between the topics of materials and methods. Renaming this section as "Methodology" and creating distinct subsections within it would enhance its coherence and comprehensibility. This particular section demands considerable enhancement from the authors to provide a clearer and more organized framework.

The conclusion appears somewhat fragile. Strengthening it involves incorporating discussions on the study's limitations, along with its theoretical and practical contributions. This addition will fortify the conclusion and provide a comprehensive overview of the study's implications.  They need to be clearer and more specific. Also add the limitations of the study.

To bolster the background, additional references are needed to support the discussion. Both circular economy and artificial intelligence have extensive literature, and integrating recent theories and studies on these subjects would significantly augment the paper's depth and credibility

Round 2

Reviewer 2 Report

Comments and Suggestions for Authors

Accept in present form

Comments on the Quality of English Language

Accept in present form

Author Response

First of all, we would like to thank the editor for his time in reviewing the manuscript. The comments were very helpful in revising and improving our manuscript. As noted by the editor, we have added comments on the importance of the research, justification of the methodology, added citations in various sections, and revised the conclusions to elaborate on the implications and contributions.

Reviewer 3 Report

Comments and Suggestions for Authors

Now, the manuscript appears to have been revised, and can proceed to the publication process.

Author Response

(The authors gave the same response as above.)
